# VC Theoretical Explanation of Double Descent

## Abstract

There has been growing interest in generalization performance of large multilayer neural networks that can be trained to achieve zero training error, while generalizing well on test data. This regime is known as 'second descent' and it appears to contradict the conventional view that optimal model complexity should reflect an optimal balance between underfitting and overfitting, i.e., the bias-variance trade-off. This paper presents a VC-theoretical analysis of double descent and shows that it can be fully explained by classical VC-generalization bounds. We illustrate an application of analytic VC-bounds for modeling double descent for classification, using empirical results for several learning methods, such as SVM, Least Squares, and Multilayer Perceptron classifiers. In addition, we discuss several reasons for the misinterpretation of VC-theoretical results in Deep Learning community.

## 1 Introduction

There have been many recent successful applications of Deep Learning (DL). However, at present, various DL methods are driven mainly by heuristic improvements, while theoretical and conceptual understanding of this technology remains limited. For example, large neural networks can be trained to fit available data (achieving zero training error) and still achieve good generalization for test data. This contradicts the conventional statistical wisdom that overfitting leads to poor generalization. This phenomenon has been systematically described by Belkin et al. (2019) who introduced the term 'double descent' and pointed out the difference between the classical regime (first descent) and the modern one (second descent). The disagreement between the classical statistical view and modern machine learning practice provides motivation for new theoretical explanations of the generalization ability of DL networks and other over-parameterized estimators. Several different explanations include: special properties of multilayer network parameterization (Bengio, 2009), choosing proper inductive bias during second descent (Belkin et al., 2019), the effect of Stochastic Gradient Descent (SGD) training (Zhang et al., 2021; Neyshabur et al., 2014; Dinh et al., 2017), the effect of various heuristics (used for training) on generalization (Srivastava et al., 2014), and the effect of margin on generalization (Bartlett et al., 2017). The current consensus view on the 'generalization paradox' in DL networks is summarized below:

- Existing indices for model complexity (or capacity), such as VC-dimension, cannot explain generalization performance of DL networks.
- 'Classical' theories developed in ML and statistics cannot explain generalization performance of DL networks and the double descent phenomenon. Specifically, Zhang et al. (2021) argues that the ability of large DL networks to achieve zero training error (during second descent mode) effectively "rules out all of the VC-dimension arguments as a possible explanation for the generalization performance of state-of-the-art neural networks."

This paper demonstrates that these assertions are incorrect, and that classical VC-theoretical results can fully explain generalization performance of DL networks, including double descent, for classification problems. In particular, we show that proper application of VC-bounds using correct estimates of VC-dimension provides accurate modeling of double descent, for various classifiers trained using SGD, least squares loss and standard SVM loss. The proposed VC-theoretical explanation provides additional insights on generalization performance during first descent vs. second descent, and on the effect of statistical properties of the data on double descent curves.

Next, we briefly review VC-theoretical concepts and results necessary for understanding generalization performance of all learning methods based on minimization of training error (Vapnik, 1998; 1999; 2006; Cherkassky & Mulier, 2007):

1. Finite VC-dimension provides *necessary* and *sufficient* conditions for good generalization.
2. VC-theory provides analytic bounds on (unknown) test error, as a function of training error, VC-dimension and the number of training samples.

Clearly, these VC-theoretical results contradict an existing consensus view that VC-theory cannot account for generalization performance of large DL networks. This disagreement results from a misinterpretation of basic VC-theoretical concepts in DL research. These are a few examples of such misunderstanding:

– A common view that VC-dimension grows with the number of parameters (weights), and therefore, "traditional measures of model complexity struggle to explain the generalization ability of large artificial neural networks" (Zhang et al., 2021). In fact, it is well known that VC-dimension can be equal, or larger, or smaller, than the number of parameters (Vapnik, 1998; Cherkassky & Mulier, 2007).

– Another common view is that "VC-dimension depends only on the model family and data distribution, and not on the training procedure used to find models" (Nakkiran et al., 2021). In fact, VC-dimension does not depend on data distribution (Vapnik, 1998; Cherkassky & Mulier, 2007; Schölkopf & Smola, 2002). Furthermore, VC-dimension certainly depends on SGD algorithm (Vapnik, 1998; Cherkassky & Mulier, 2007).

For classification problems, VC-theory provides analytic generalization bounds for (unknown) Prediction Risk (or test error $R_{tst}$), as a function of Empirical Risk (or training error $R_{trn}$) and VC-dimension ($h$) of a set of admissible models. That is, for a given training data set (of size $n$), VC-bound has the following form (Vapnik, 1998; 1999; 2006; Cherkassky & Mulier, 2007):

$$R_{tst} \leq R_{trn} + \frac{\varepsilon}{2} \left( 1 + \sqrt{1 + \frac{4R_{trn}}{\varepsilon}} \right), \text{ where } \varepsilon = \frac{a_1}{n} \left( h \left( \ln \left( \frac{a_2 n}{h} \right) + 1 \right) - \ln \frac{\eta}{4} \right) \quad (1)$$

This VC-bound (1) holds with a probability of $1 - \eta$ for all possible models (functions) including the one minimizing $R_{trn}$. The value of $\eta$ is preset to a small value, i.e., $\eta = 4/\sqrt{n}$. The second additive term in (1), called the *confidence interval* (also known as *excess risk*), depends on both $R_{trn}$ and $h$. This bound describes the relationship between training error, test error, and VC-dimension, and it is used for conceptual understanding of model complexity control, i.e. the effect of VC-dimension on test error. Application of this bound for accurate modeling of double descent curves requires:

– *Selecting proper values of positive constants $a_1$ and $a_2$.* The worst-case values $a_1 = 4$ and $a_2 = 2$, provided in VC-theory (Vapnik, 1998; 1999) correspond to the worst-case "heavy-tailed" distributions, resulting in VC-bounds that are too loose for real-life data sets (Cherkassky & Mulier, 2007). For real-life data sets, when distributions are unknown, we suggest the values $a_1 = 3$ and $a_2 = 1$, that were used for all empirical results in this paper. For additional discussion about selecting these values (incorporating *a-priori* knowledge about unknown distributions), see Appendix A.

– *Analytic estimates of VC-dimension.* For many learning methods (including DL), analytic estimates of VC-dimension are not known. For example, for SGD-style algorithms, the effect of various heuristics (e.g., initialization of weights, etc.) on VC-dimension is difficult (or impossible) to quantify analytically.

Note that VC-bound (1) provides a conceptual explanation of both first and second descent. That is, first descent corresponds to minimizing this bound when training error is non-zero (Vapnik, 1998; 1999; Cherkassky & Mulier, 2007). Second descent corresponds to minimizing this bound when training error is kept at zero, using models having small VC-dimension. This can be shown by setting the training error in bound (1) to zero, resulting in the following bound for test error during the second descent (uisng values $a_1 = 3$ and $a_2 = 1$):

$$R_{tst} \leq \varepsilon, \qquad \text{where } \varepsilon = \frac{3h}{n} \left( \ln \left( \frac{n}{h} \right) + 1 \right) \quad (2)$$

Technically, since VC-bound (1) depends only on two factors, training error and VC-dimension, there are two different strategies for minimizing this bound (Cherkassky & Mulier, 2007):

– *Strategy 1*: for a set of functions (models) with fixed VC-dimension, reduce the training error. This leads to well-known classical bias-variance trade-off, also known as first descent;

– *Strategy 2*: for small (fixed) training error, minimize the VC-dimension. This strategy corresponds to second descent.

These are two different strategies for controlling VC-dimension, leading to different implementations of Structural Risk Minimization (SRM) in VC-theory. Strategy 2 is formally described in VC-theory by considering a set of functions (models) where the training error remains constant, called an *equivalence class* (Vapnik, 1998; 1999; Cherkassky & Mulier, 2007). For example, all models during second descent form an equivalence class (since the training error is zero). For each equivalence class, we define a structure, or complexity ordering, according to the *norm squared* of weights (of a linear classifier). Then an optimal model (minimizing the norm squared of weights) is found using training data. Such a strategy has been implemented via maximization of margin in SVM classifiers. Most learning methods typically implement a *single strategy*, whereas practitioners in DL observed the effect of *both strategies* when varying a single hyper-parameter, such as network size or the number of epochs. Therefore, double descent curve has a simple VC-theoretical explanation. For instance, consider the effect of increasing the number of hidden units N (in a single-layer network) on the test error. When the number of hidden units is small, SGD training aims to minimize training error, and VC-dimension is increasing with the number of hidden units. This corresponds to Strategy 1 for minimizing the VC-bound (1). On the other hand, for over-parameterized networks (large N), SGD training finds a minimum norm of weights solution - effectively implementing Strategy 2 for minimizing VC bound (1). The mystery of improved generalization performance during second descent is explained by noting that for larger networks, VC-dimension is actually decreasing. This is confirmed by the empirical modeling results presented later in Section 2.

Another technical reason for misapplication of VC-theory, besides misinterpretation of VC-dimension, is that VC-bound (1) remains virtually unknown in the DL community. Many papers suggesting that VC-theory is unable to explain double descent, are based on analysis of *uniform convergence bounds* (Belkin et al., 2019; Zhang et al., 2021; Tewari & Bartlett, 2014; Bartlett et al., 2021; Koltchinskii, 2001; Sokolić et al., 2017). In such bounds, the confidence interval term (or excess error), is of the order $\mathcal{O}\left(\sqrt{h/n}\right)$. However, VC-theory also provides a more accurate *uniform relative convergence* bounds, such as VC-bound (1), presented in (Vapnik, 1998; 1999; 2006; Cherkassky & Mulier, 2007), where the excess error is of the order $\mathcal{O}\left(h/n\right)$. For small values of $h/n$, common in practical applications, this difference is large. For example, for $h/n = 0.1$, uniform relative convergence bounds give $\sim 10\%$ excess error, but the uniform convergence bounds give $\sqrt{0.1}$ or $\sim 33\%$. So, while it is true that uniform convergence bounds cannot accurately model second descent, it can be done using uniform relative convergence bounds.

Training of DL networks is based on SGD, which incorporates several heuristic rules to ensure that the norm of weights remains small. These rules include: initialization of weights to small random values, weight decay, and re-normalization during training. Consequently, for large networks, model complexity is determined by the norm of weights, rather than the number of weights (parameters). Further, this dependence of VC-dimension on the training algorithm helps explain why theoretical estimates of VC-dimension based on network topology (network size) have found little practical use (Cherkassky & Mulier, 2007).

## 2 APPLICATION OF VC BOUNDS FOR MODELING DOUBLE DESCENT

This section presents a VC-theoretical explanation of double descent for classification, for a single-layer network shown in Figure 1. The same network setting was used for analysis of double descent in recent papers (Belkin et al., 2019; 2020; Neal et al., 2018; Hastie et al., 2019). In this network, a classifier is estimated in two steps:

– First, input vector $x$ is encoded using N nonlinear features in $\mathbf{Z}$-space. Commonly, *random features* (weak features) are used, such as random ReLU or Random Fourier Features (RFF);

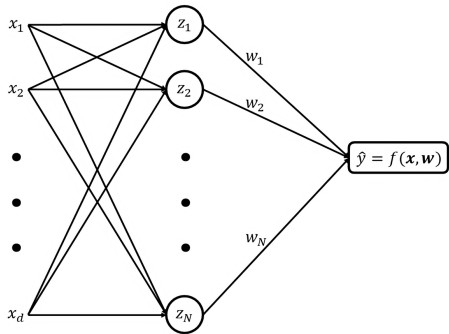

Figure 1: A single hidden layer network estimating a linear classifier in nonlinear feature space.

— Second, a linear model is estimated in this N-dimensional feature space.

This simplified setting enables VC theoretical analysis of double descent, because the analytic estimates of VC-dimension are known. That is, since the network output is formed as a linear combination of N features, the analytic estimate of VC-dimension for linear hyperplanes $f(\boldsymbol{z}, \boldsymbol{w}) = (\boldsymbol{w}\boldsymbol{z}) + b$ is known (Vapnik, 1998; 1999; 2006):

$$h \leq \min\left(||\boldsymbol{w}||^2, N\right) + 1 \qquad (3)$$

This bound holds under the assumption that all training samples are enclosed within a sphere of radius 1, in $\boldsymbol{Z}$-space. In summary, VC-dimension can be bounded by the input dimensionality (N), or by the norm squared of weights $\boldsymbol{w}$. These are two different mechanisms for controlling VC-dimension. In machine learning literature, the expression (3) is widely known as scale-sensitive or fat-shattering VC-dimension (Schölkopf & Smola, 2002; Shawe-Taylor et al., 1996), where shattering is performed by delta margin hyperplanes (with constraints on margin size or $||\boldsymbol{w}||^2$).

The double descent phenomenon can be observed for various learning methods used to estimate weights $\boldsymbol{w}$ for the network structure in Figure 1. Next, we present empirical results showing the application of VC-bounds (1) and (3) for modeling double descent when network weights are estimated using SVM or Least-Squares (LS) classifiers. For large networks trained using LS, when N is larger than sample size (*n*), minimization of squared error is performed using pseudo-inverse, which finds a solution corresponding to the minimization of the norm squared $||\boldsymbol{w}||^2$.

In all experimental results in this section, double descent is observed when the network size (N) is gradually increased. Specifically, according to analytic bound (3):

— When network size (N) is small, the VC-dimension initially grows linearly with N. This corresponds to the first descent, or traditional bias-variance trade-off.

— For overparameterized networks (large N), VC-dimension is controlled by the norm squared of weights, leading to second descent.

We use two types of random nonlinear features (Belkin et al., 2019; Rahimi & Recht, 2008), ReLU and RFF. Random ReLU features are formed as:

$$\boldsymbol{Z}_i = \max\left(\langle \boldsymbol{v}_i, \boldsymbol{X}\rangle, 0\right), \quad i = 1, ..., N$$

where random vectors $\boldsymbol{v}_1, \ldots, \boldsymbol{v}_N$ are sampled uniformly from the range [-1,1]. Random Fourier Features (RFF) are formed as:

$$\boldsymbol{Z}_i = \exp\left(\sqrt{-1}\langle \boldsymbol{v}_i, \boldsymbol{X}\rangle\right), \quad i = 1, ..., N$$

Where random $\boldsymbol{v}_1, \ldots, \boldsymbol{v}_N$ are sampled from Gaussian distribution with standard deviation $\sigma = 0.05$. In all experiments, input ($\boldsymbol{x}$) values were pre-scaled to [0, 1] range, for training and test data.

Following the nonlinear mapping $\boldsymbol{X} \to \boldsymbol{Z}$, all $\boldsymbol{z}$-values are re-scaled to [-1, 1] range. Such rescaling is performed to satisfy the condition for bound (3), stating that all training samples in $\boldsymbol{Z}$-space should be enclosed within a sphere of radius 1.

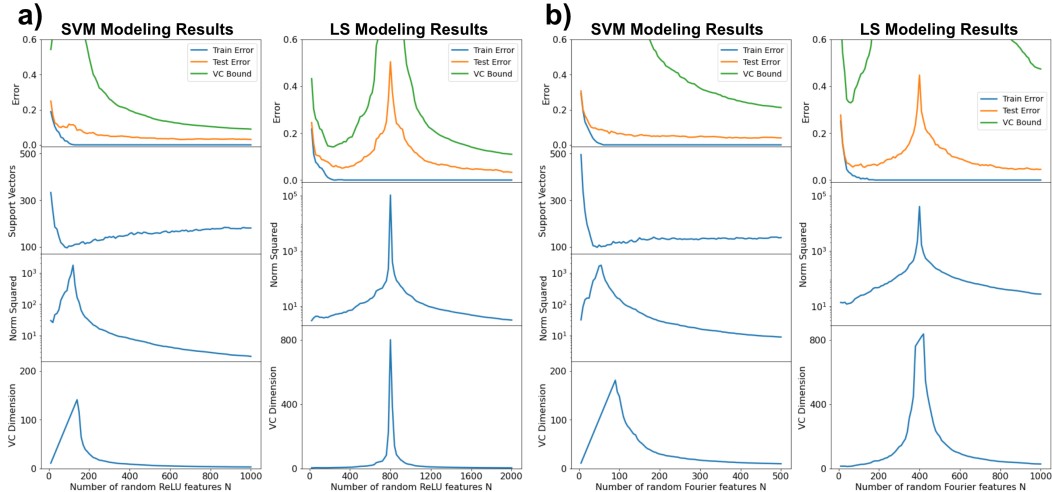

Figure 2: Application of VC-bounds for MNIST digit 5 vs 8 data set using **a)** random ReLU features and **b)** RFF.

Training samples $(\boldsymbol{z}, y)$ are used to estimate a decision boundary in **Z**-space. Two different methods (LS and SVM classifiers) are used for estimating linear decision function $f(\boldsymbol{z}, \boldsymbol{w}) = (\boldsymbol{wz}) + b$ from training data, in order to show double descent curves for two *different loss functions*, LS and SVM loss. For SVM modeling, the regularization parameter $C$ is set to 64 in all experiments.

Most empirical results in this paper were obtained for MNIST digits adapted for binary classification (digit 5 vs 8), where digits are grey-scale images of size 28x28. The training set size is $n = 800$ and test set size is 2,000. We have also used other data set for modeling and observed similar results. See Appendix A for additional results.

Figures 2(a) and 2(b) show application of VC-bounds to modeling MNIST data. They show:

- Empirical training and test error curves, as a function of N (the number of nonlinear features), along with VC-theoretical estimate of test error obtained via bounds (1) and (3). These curves show that analytic VC-bounds can explain (and predict) double descent;
- The norm squared of weights of estimated linear models, as a function of the number of features N;
- The estimated VC-dimension via bound (3), as a function of the number of features N;
- For SVM, we also show the number of support vectors for trained SVM model.

Modeling results for random ReLU features and RFF are similar, so we only comment on results in Figure 2(a):

- *For small N*, VC-dimension grows linearly with N for SVM method. Empirical results show that first descent error curves can be explained by VC-bound (1), because the minimum of VC-bound closely corresponds to the minimum of test error. This can be clearly seen for LS classifier, but less obvious for SVM.
- *For large N*, VC-dimension is controlled by the norm squared, according to bound (3). These results show that second descent can be explained by VC-bound (1), for both SVM and LS learning methods.

Whereas empirical results for both SVM and LS in Figure 2(a) are qualitatively similar, their double descent curves show different values of *interpolation threshold* N* (where the training error reaches zero). For SVM, the value N* $\approx$ 100 is achieved when the number of features equals the number of support vectors. For LS classifier, the interpolation point N* $\approx$ 800 is achieved when the number of features equals the number of training samples.

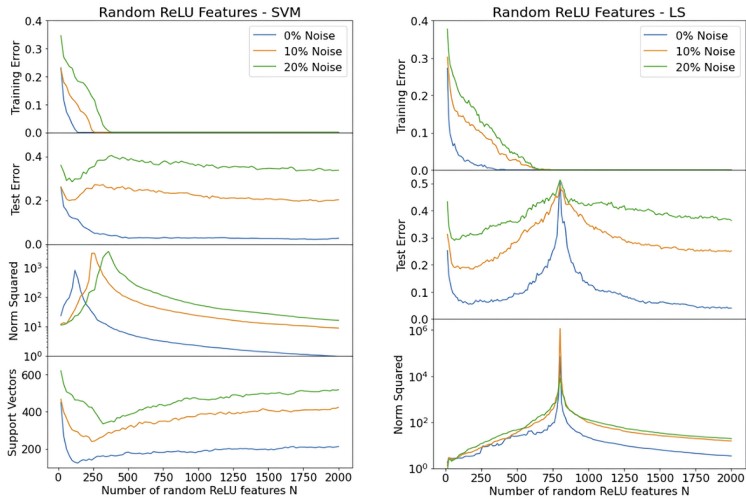

Figure 3: Effect of noise on double descent for MNIST data, for SVM and LS classifiers using random ReLU features.

The dependence of test error on the norm of weights in large networks has been known to practitioners, and some limited theoretical explanation is provided in (Belkin et al., 2019; 2018; Gunasekar et al., 2017). For example, Belkin et al. (2019; 2018) suggest that minimum norm provides inductive bias by favoring models with a higher degree of smoothness. However, these papers do not mention VC-bounds that clearly relate the VC-dimension to the norm of weights, and explain generalization performance for linear classifiers. Our results also show that the same VC-bounds apply with different loss functions used for training, i.e., SVM and LS loss. For both loss functions, second descent implements the same VC-theoretical structure where the elements of a structure are ordered according to the norm squared of weights. For large N, the estimated model trained using SVM loss approaches a standard kernel SVM solution, whereas a model trained using LS loss approaches a Least-Squares SVM solution (Suykens & Vandewalle, 1999).

The dependence of interpolation threshold N* on training sample size for LS classifiers has also been observed in the DL literature. However, in the absence of sound theoretical framework for double descent, interpretation of this empirical dependency leads to convoluted explanations. For example, Nakkiran et al. (2021) investigated the effect of varying the number of training samples on test error, for a fixed-size DL network. In particular, they observed two double descent error curves for the same network trained using smaller and larger size training data. Under this setting, near interpolation threshold, the test error for a network trained with a larger data set is worse than for the same network trained on a smaller data set. This phenomenon was called 'sample-wise non-monotonicity', and a new theory was proposed for explaining regimes where 'increasing the number of training samples actually hurts test performance'. However, this phenomenon has a simple VC-theoretical explanation, presented next. During second descent, the shape of the 'norm squared' closely follows the shape of test error, according to VC-bound (2), as evident in Figures 2a and 2b. Since for LS classifiers the interpolation threshold is given by training size, there is a region near interpolation threshold where VC-dimension for a smaller training size is smaller than for a larger training size. So, in this region we can expect a smaller test error for smaller training size.

VC-theoretical framework can also help to understand the effect of statistical characteristics of training data on generalization curves. Next, we present empirical results demonstrating the effect on noisy data on the shape of double descent curves, along with their VC-theoretical explanation. For these experiments, we use a single-layer network trained using SVM and LS classifiers using random ReLU features. We use digits data with randomly corrupted class labels. The training set size is 800 (400 per class), and the test set size is 2,000. Figure 3 shows the effect of noise level on the shape of double descent curves, for SVM and LS classifiers. Results for both SVM and LS models show double descent curves, but their shape is different. For the SVM model estimated using 'clean' data (0% label noise), there is no visible first descent at all, but for noisy data we observe both first and second descent. For the LS model, we clearly observe first and second descent for both clean

and noisy training data. For LS curves, the interpolation threshold ($\approx$ training size 800) is the same for different noise levels, but for SVM the value of interpolation threshold increases with noise level in the data. This can be explained by noting that for SVM, the interpolation threshold is reached when the training data becomes linearly separable (in nonlinear feature space, or $\mathbf{Z}$-space in Figure 1). Therefore, for SVM the interpolation threshold is given by the number of support vectors needed to separate training data. For noisy data, a SVM model requires a larger number of support vectors, resulting in larger interpolation threshold.

The ability to generalize for noisy data can be explained by noting that during second descent VC-bound (2) depends only on VC-dimension (the norm of weights). With increasing noise (in the data), the norm of weights increases, resulting in degradation of test error and flattening of second descent test error curve (as evident in Figure 3, for both SVM and LS).

Empirical results in Figure 3 also show that for noisy data, generalization performance during second descent degrades relative to an optimal first descent model. This is contrary to the popular view that DL networks usually provide superior generalization performance during second descent (Belkin et al., 2019; Zhang et al., 2021; Neyshabur et al., 2014; Nakkiran et al., 2021).

We suspect that superior performance during second descent, reported in the DL community, can be explained by using large and 'clean' data sets (common in Big Data). For such training data sets (of large size $n$), generalization performance during second descent is likely to be good, because the VC-bound (2) on test error depends only on the ratio of VC-dimension to sample size ($h/n$).

## 3  MODELING DOUBLE DESCENT FOR MULTILAYER NETWORKS

Empirical results for a simplified network setting in Section 2 provide insights for generalization performance of over-parameterized multilayer networks. Such general DL networks use SGD training that keeps the norm of weights small, so that the model complexity is determined by the norm of weights, rather than the number of weights. However, direct application of analytic VC-bounds to modeling double descent may be tricky, due to two challenging research issues:

1. How to estimate VC-dimension for DL networks, where analytic estimates do not exist;

2. Understanding design choices for setting multiple 'tuning' parameters, such as network width, number of training epochs, weight initialization, etc. All of these hyperparameters can be used to control the VC dimension of DL networks. Double descent curves show dependence of training and test error on a *single complexity parameter*, when all other tuning parameters are preset to reasonably 'good' values.

For these reasons, the application of analytic VC-bounds to general DL networks is difficult. However, it can still be done for restricted and well-defined network settings. In this section, we consider a fully connected network with a single hidden layer, as in Figure 1, where the network weights in *both layers* are estimated during training via SGD. In this case, $z$-features are *adaptively estimated* from training data, in contrast to *fixed* random features used earlier in Section 2.

Let us consider two factors (hyperparameters) controlling the complexity of such networks trained via SGD: the number of hidden units (N), and the number of training epochs. Empirical results showing double descent curves as a function of these two factors have been extensively reported in DL literature (Belkin et al., 2019; Nakkiran et al., 2021). However, our purpose here is not to replicate such double descent curves, but to explain them using VC-bounds (1) and (3). In order to achieve this, we specify a network settings where the VC-dimension can be approximately estimated. Therefore, we only consider over-parameterized (wide) networks during second descent, i.e., when the number of hidden units N is large, or the number of training epochs is large. We hypothesize that under *such restricted settings*, the VC-dimension of a neural network can be estimated as the norm squared of weights of the output layer. This hypothesis is based on the recent understanding that during second descent regime *sufficiently wide single-layer* networks (trained via SGD) closely approximate a linear classifier shown in Figure 1. See Belkin (2021) for a mathematical explanation of this phenomenon called 'transition to linearity'. Similarly, Bietti & Bach (2021) showed that wide fully-connected multilayer networks have essentially the same approximation properties as their "shallow" two-layer counterparts, implementing standard kernel machines.

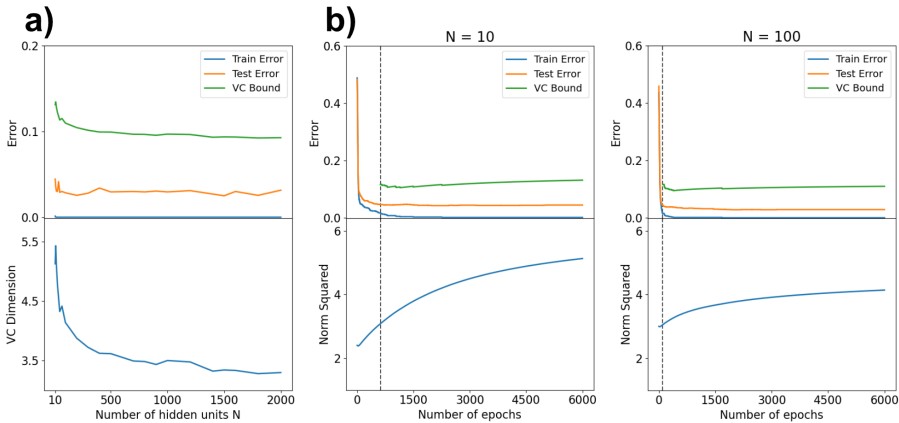

Figure 4: VC-theoretical modeling of second descent in a fully connected network. **a)** Second descent as a function of network width (N). **b)** Second descent as a function of the number of epochs, for N = 10 and N = 100.

According to this interpretation, minimization of the norm of weights in the output layer is mainly responsible for generalization, whereas all previous layers perform non-linear transformation of input **x**, by encoding it as a large number of weak nonlinear features ($\sim$ N nonlinear features in **Z**-space). We argue that this assumption is plausible for data sets originating from well-separable (low-noise) class distributions, such as MNIST digits – where good generalization can indeed be achieved during second descent, by both linear and nonlinear networks (of large width). Therefore, we restrict modeling of second descent for nonlinear networks using VC-bounds, presented in this section, only for such separable distributions.

Our experiments show application of VC-bound (2) *under such restricted settings*, i.e., for over-parameterized fully connected networks trained using SGD, using the norm of weights in the output layer as an estimate of VC-dimension during second descent.

We present empirical results for MNIST digits data, under the following experimental setting:

- 800 training and 2,000 test examples (of digits 5 and 8);
- Fully connected network using ReLU activation function in hidden units (Paszke et al., 2019);
- Training using SGD with learning rate 0.001 and momentum 0.95. The learning rate is reduced by 10% for every 500 epochs. Batch normalization is used during training.
- Weights initialized, prior to training, using Xavier uniform distribution, following Glorot & Bengio (2010).

Our design choices for SGD implementation mainly follows earlier studies (Belkin et al., 2019; Glorot & Bengio, 2010).

Figure 4(a) shows modeling results for second descent mode, as a function of the number of hidden units N (the number of epochs is set to 6000 in all experiments). These results show empirical training and test error curves, and the VC-bound on test error, that closely approximates empirical test error. This figure also shows the VC-dimension, estimated as the norm squared of the output layer weights. It is interesting to compare modeling results in Figure 4(a) (for nonlinear network) and LS results shown in Figure 2(a) for the same data set, but using a linear network. First, we can observe similarly decreasing norm squared of the output layer weights, in both figures. Second, direct comparison of the norm squared of weights (of the output layer), for linear and nonlinear models, for large networks, indicates their similar values. For network size N= 2,000, the norm squared of the weights (and test error) equals 3.3 (3.7%) for a linear network and 2.3 (3.1%) for a nonlinear network. Since the norm squared of weights was used as VC-dimension in bound (2), for both linear and nonlinear networks, these comparisons support our hypothesis about using this norm squared, as a proxy for VC-dimension during second descent.

Figure 4(b) shows modeling results for second descent, as a function of the number of epochs (for networks with N = 10 and 100 hidden units). Note that VC-dimension can be reliably estimated only in second descent mode, when training error is close to zero. This region, where training error is under 1%, is indicated by dotted vertical line. In this region, increasing the number of epochs results in a small increase in VC-dimension and a slight decrease of training error. This is a particular form of memorization-complexity trade-off, implicit in VC-bound (1), when training error is close to zero. For additional results on modeling double descent in multilayer networks, see Appendix B.

These results demonstrate application of VC-bounds for modeling second descent in multilayer networks. In addition, we can see the effect of each complexity parameter (network size N and the number of epochs) on VC-dimension. This can be used for ranking tuning parameters, according to their ability to control VC-dimension of DL networks during second descent.

## 4   SUMMARY AND DISCUSSION

This paper provides a VC-theoretical explanation of 'double descent' in multilayer networks. We show that for simplified network settings, where analytic estimates of VC-dimension exist, VC-generalization bounds can be applied directly to predict double descent phenomenon. VC-theoretical framework is helpful for improved understanding of empirical results observed in DL, such as: modeling of double descent, the effect of various heuristics on generalization, comparing generalization during first and second descent, etc. According to VC-theoretical explanation, second descent occurs when zero training error is achieved using an estimator having small norm of weights. This phenomenon is general, and it does not depend on particular training algorithm or chosen model parameterization. Therefore, double descent can be observed for other learning methods, such as SVM estimators.

VC theoretical interpretation of double descent has several methodological implications. As explained in this paper, the first and second descent modes of learning implement two different types of VC-theoretical 'structures', or complexity orderings, of possible models. Both types of structures have been well-known, but are usually presented separately when analyzing different learning methods (Vapnik, 1998; 1999; Cherkassky & Mulier, 2007). Second descent mode of learning has been studied in the past, long before DL, albeit under different terminology, such as margin maximization in standard SVM, Least-Squares SVM classifiers (Suykens & Vandewalle, 1999), and Optimal Linear Associative Memory (Kohonen, 1989). Recently, the similarity between second descent and kernel learning has been rediscovered by several authors (Belkin et al., 2018; Belkin, 2021; Bietti & Bach, 2021).

According to VC-methodology, we should analyze the first and second descent modes of learning *separately*, as they implement two different types of SRM structures. Just showing test error curves for these two structures on the same figure, as double descent, does not provide much technical insight. An important open question is understanding under what conditions a model estimated during first descent provides better generalization than a model estimated during second descent. Current DL literature suggests that second descent provides better generalization performance, for most 'real-life' data sets – but this statement requires better technical explanation.

According to VC-theoretical explanation, superior generalization performance during second descent can be expected for large sample size, that are inherently separable, i.e. have small Bayes optimal test error (say, under 5%). In this case, using a structure based on the norm of weights results in a second descent mode, where training error is zero and VC-bound on test error is given by (2). According to this expression (2), this bound is expected to be small, for small values of the ratio of VC-dimension to sample size ($h/n$).

Finally, we point out that VC-theoretical explanation of double descent utilizes just a few well known concepts, such as VC-dimension and Structural Risk Minimization. The same VC-theoretical concepts and results explain generalization performance of other methods, such as parametric estimators, SVM, signal denoising, etc (Cherkassky & Mulier, 2007). This is in contrast to multiple recent theories for explanation of double descent, that introduce various new theoretical constructs and associated technical jargon, aimed to explain this particular empirical phenomenon.

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

## APPENDIX A: SELECTION OF PARAMETERS IN VC-BOUNDS

This Appendix includes additional empirical results for modeling double descent in a single-layer network shown in Figure 1 in the main paper. First set of results investigates the choice of theoretical constants $a_1$ and $a_2$ in VC-bound (1) reproduced below:

$$R_{tst} \leq R_{trn} + \frac{\varepsilon}{2}\left(1 + \sqrt{1 + \frac{4R_{trn}}{\varepsilon}}\right)$$

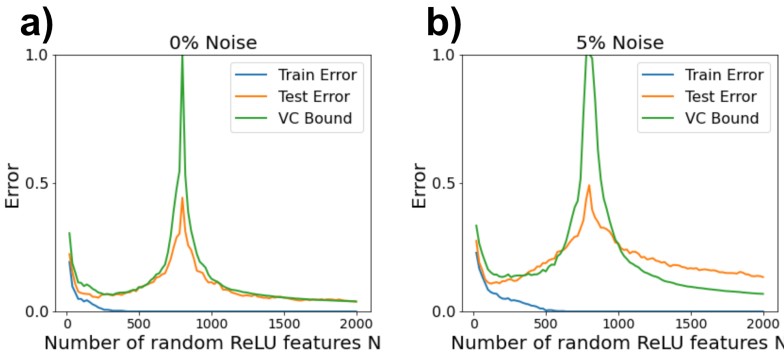

Figure 5: Modeling double descent for MNIST data set (digit 5 vs 8) using values $a_1 = 1$ and $a_2 = 1$, with **a)** original data set (no label noise) and **b)** 5% label noise added.

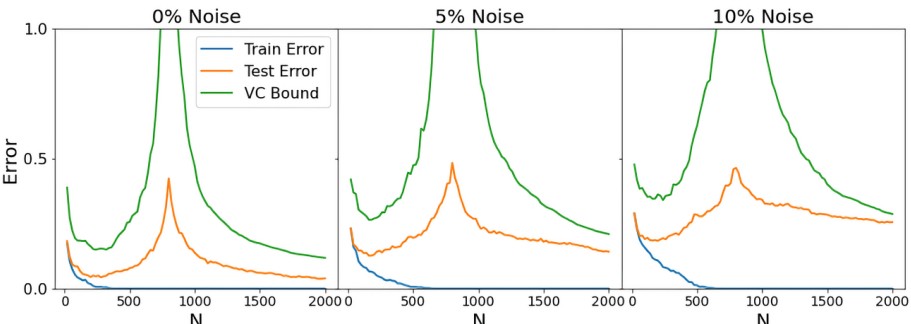

Figure 6: Modeling double descent for digits data with corrupted class labels, using values $a_1 = 3$ and $a_2 = 1$.

$$\text{where } \varepsilon = \frac{a_1}{n} \left( h \left( \ln \left( \frac{a_2 n}{h} \right) + 1 \right) - \ln \frac{\eta}{4} \right), \eta = \frac{4}{\sqrt{n}}$$

VC-theory (Vapnik, 1998; 1999) specifies their range and provides the values corresponding to pessimistic assumptions (about unknown data distributions):

- the range [0, 4] for $a_1$ and [0,2] for $a_2$.
- worst-case values $a_1 = 4$ and $a_2 = 2$.

These worst-case values result in upper bounds that are too crude for real-life data sets. It may be worth noting here that VC-bounds (and most other analytic results in VC-theory) are regarded as conceptual, and they have not been used for modeling real-life data sets. However, in this paper, we apply VC-bounds for modeling double descent, so selecting 'practical' values of these constants becomes critical. The original reference on VC-theory (Vapnik, 2006) provides a technical discussion on the problem of incorporating prior knowledge (about unknown distributions) into VC-bounds. Whereas *a-priori* knowledge about particular form of distribution allows to obtain accurate results (for that distribution), such results have limited practical value. Therefore, we can only provide general guidelines for selecting proper values for values $a_1$ and $a_2$, based on empirical results for multiple data sets. Such guidelines, along with their empirical justification, are presented next.

The overall recommendation is that using values $a_1 = 3$ and $a_2 = 1$ results in practical VC-bounds providing accurate and robust modeling of double descent for real-life data sets. For 'clean' data sets from well-separable class distributions using values $a_1 = 1$ and $a_2 = 1$ results in more accurate VC-bounds. Applying bounds with values $a_1 = 1$ and $a_2 = 1$ to model clean and noisy data sets is illustrated in Figure 5(a), showing modeling results for digits data set used in Section 2 of the

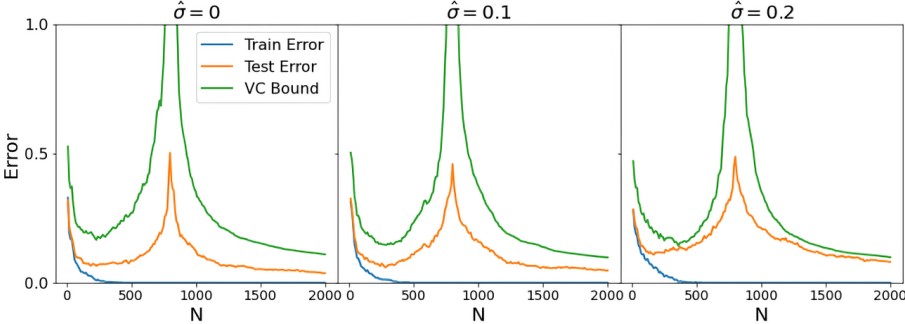

Figure 7: Modeling double descent for digits data with pixels corrupted by Gaussian noise, using values $a_1 = 3$ and $a_2 = 1$.

main paper. These results show that VC-bounds provide accurate estimates for clean data set, but underestimate empirical curves for noisy data in Figure 5(b), for both first and especially second descent. However, using suggested practical values $a_1 = 3$ and $a_2 = 1$ results in practical VC-bounds that provide accurate modeling of double descent for this data, at various noise levels. See empirical results in Figure 6.

Next, we show experimental results for the same digits data set, when images are corrupted by random Gaussian noise. Here, the noise level is given by the standard deviation of the Gaussian noise $\hat{\sigma} = 0, 0.1, 0.2$. Empirical results in Figure 7 show that VC-bounds (with values $a_1 = 3$ and $a_2 = 1$) provide accurate modeling of double descent, at various noise levels. We conclude that using values $a_1 = 3$ and $a_2 = 1$ provide robust VC-theoretical modeling of double descent for noisy data.

In the following, we present the application of VC bounds on 4 different data sets adapted for binary classification, namely CIFAR10 (cat vs automobile) (Krizhevsky & Hinton, 2009), SVHN (digit 5 vs 8) (Netzer et al., 2011), Fashion MNIST (dress vs coat) (Xiao et al., 2017), and epileptic seizure (Andrzejak et al., 2001). For the image-based dataset, each sample is first transformed into grayscale image, and then rescaled to [0, 1] interval via min-max scaling. For the epileptic seizure dataset, the data set is normalized to zero mean and unit standard deviation. We follow the same training procedures as the modeling of MNIST data set shown in section 2. All data sets has 800 training samples and 2,000 test samples. Modeling results are obtained for a network with random ReLU features, trained using both SVM and LS classifier, are shown in figures 8, 9, 10, 11.

The last set of results shows the effect of varying the number of training samples on test error, for a fixed-size network, using MNIST data set. This setting was used in Nakkiran et al. (2021) for demonstrating double descent. Results in Figure 12 show modeling double descent for digits data set, using fixed-size network with N=500 and N=1500. These results are obtained for the network with random ReLU features and are trained using a LS classifier with $a_1 = 3$ and $a_2 = 1$. They show very accurate modeling of double descent using VC-bounds, under this setting.

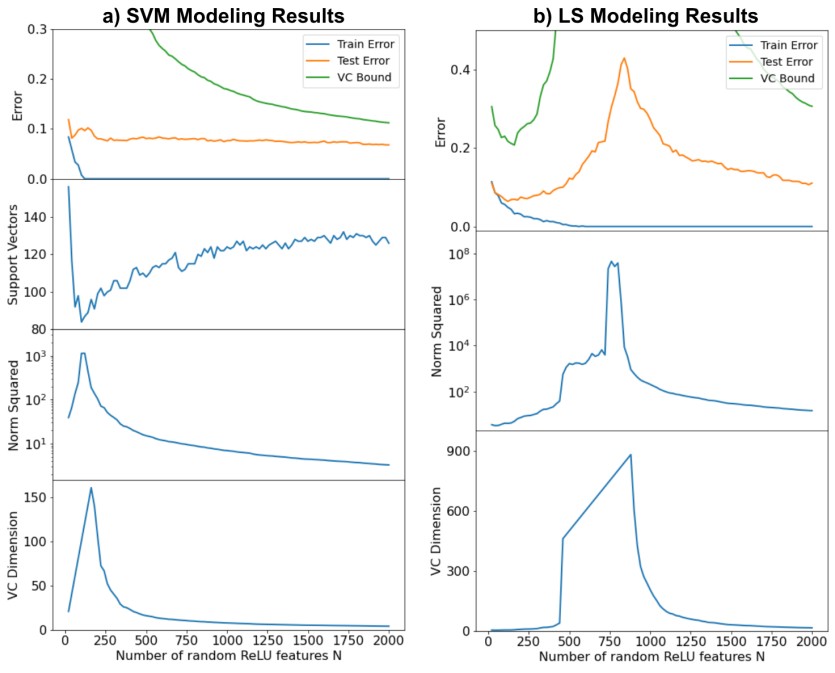

Figure 8: Application of VC-bounds for Fashion MNIST coat vs dress data set using random ReLU features.

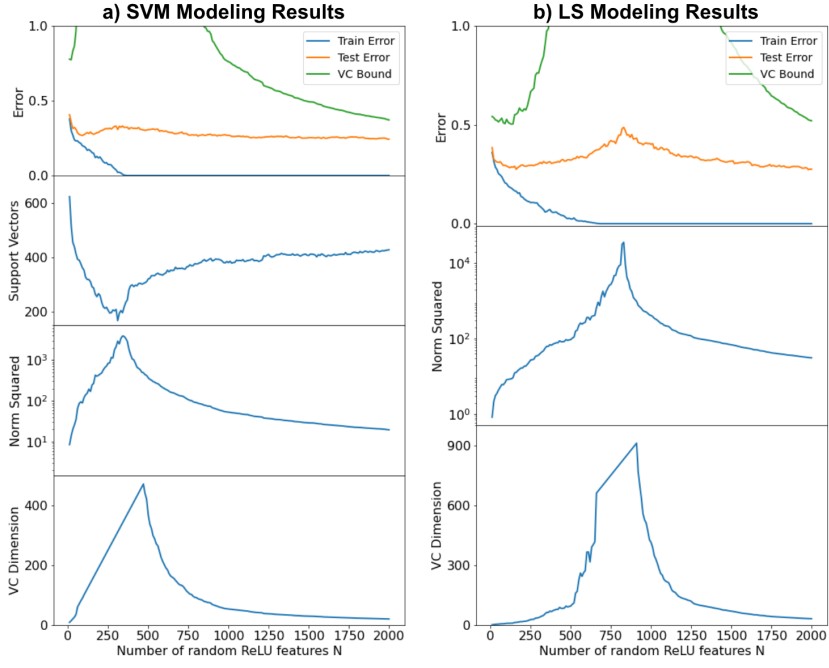

Figure 9: Application of VC-bounds for CIFAR10 cat vs automobile data set using random ReLU features.

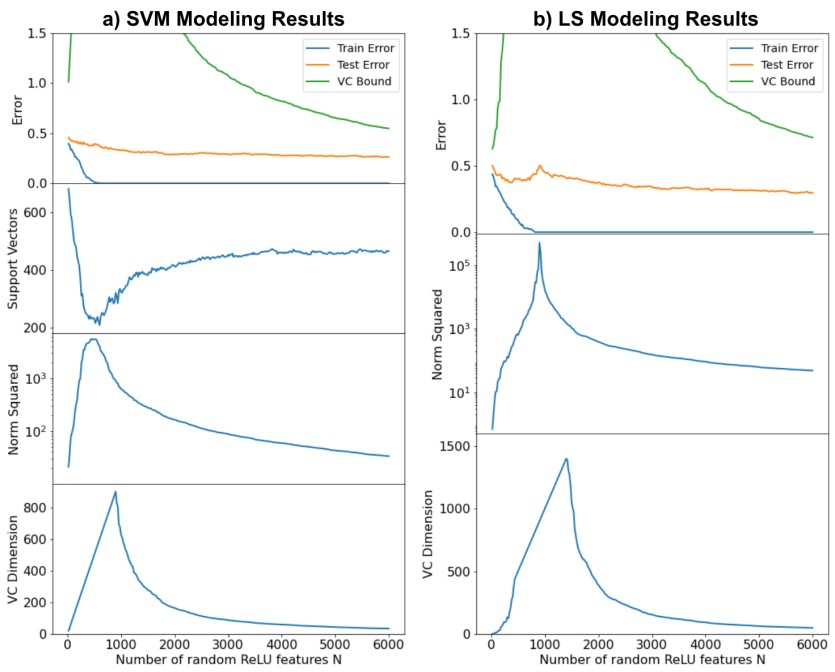

Figure 10: Application of VC-bounds for SVHN digit 5 vs 8 data set using random ReLU features.

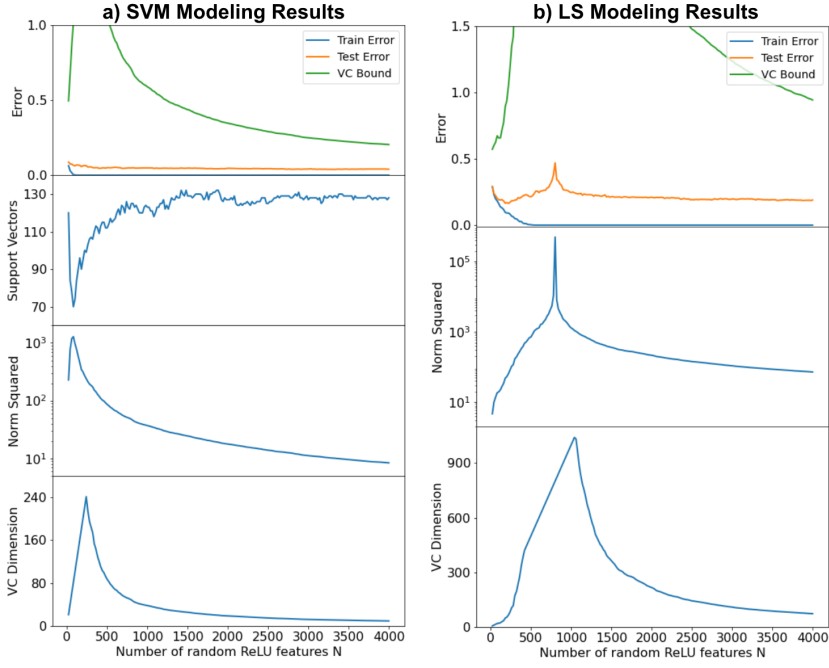

Figure 11: Application of VC-bounds for epileptic seizure data set using random ReLU features.

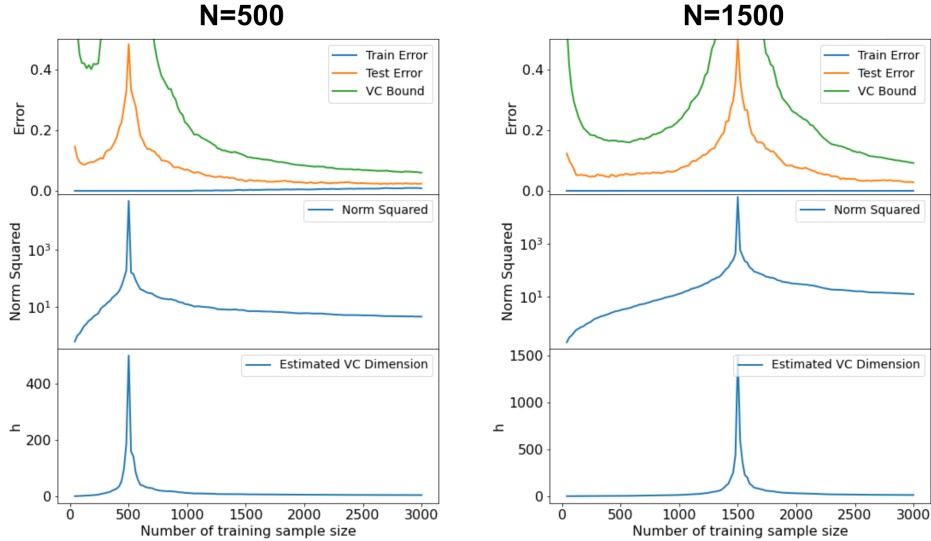

Figure 12: Modeling double descent as a function of the number of training samples, for a fixed-size network, using values $a_1 = 3$ and $a_2 = 1$.

## APPENDIX B: SECOND DESCENT MODELING FOR TWO-LAYER FULLY CONNECTED NETWORK AND CONVOLUTIONAL NETWORK

In Section 3 of the main paper, we showcase the application of VC-bounds for a 1-layer fully connected network. We considered over-parameterized networks during second descent (when training error is close to zero), and showed that, in this region: (a) VC-bounds can be applied for modeling second descent, and (b) the VC-dimension can be estimated as the norm squared of weights of the output layer. Empirical results shown in the main paper confirmed the hypothesis that in a single layer network, minimization of weights in the output layer effectively controls generalization. This appendix extends application of VC-bounds to two-layer fully connected network and convolutional LeNet-5 network (LeCun et al., 1998) for MNIST data set.

Consider a fully connected network with $N_1$ hidden units in the first layer and $N_2$ hidden units in the second layer. We adopt the same experimental setting as in the main paper. The number of epochs is set to 6000.

Figure 13 shows modeling results for a two-layer network, where $N_2$ is fixed at 10 and $N_1$ is varied. Figure 14 shows modeling results for a two-layer network, where $N_1$ fixed at 10 and $N_2$ is varied. Both Figures 13 and 14 show training and test errors similar to the results of a one-layer network in Figure 5 in the main paper. In Figures 13(b) and 14(b), the region of second descent where the training error is less than 1% is on the right side of the dotted vertical line.

Next, we present VC-theoretical modeling of second descent for a convolutional network. We use the LeNet-5 architecture (LeCun et al., 1998), modified for a binary classification problem by setting the number of output units to 1. See Figure 15 showing the schematic of the modified LeNet-5 model, where the number of hidden unit N in the last hidden layer is varied (denoted by a dark grey color). Figure 16(a) shows the error curve and VC-bound, along with the estimated VC-dimension, as a function of the number of hidden units N. Similar to multilayer network results, we observe that VC-dimension (estimated as the norm of weights) decreases as the number of hidden units N increases. Figure 16(b) shows the error curve and VC-bound, along with the norm squared of weights, as a function of the number of epochs, for N = 10 and N = 100.

Results in Figures 13, 14, and 16 confirm that VC-bounds can accurately model empirical test error, further supporting our hypothesis that VC-dimension can be estimated as the norm squared of weights of the output layer during the second descent.

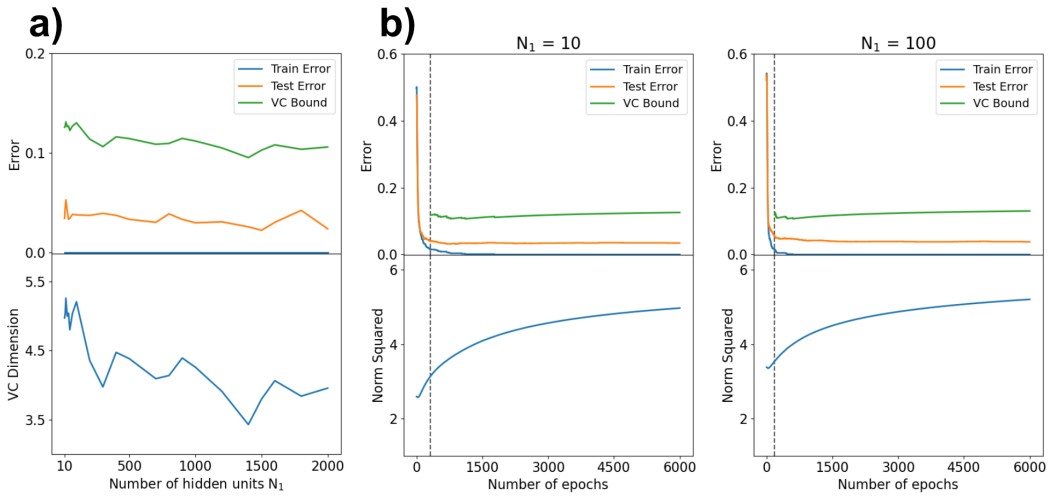

Figure 13: VC-theoretical modeling of second descent in a two-layer fully connected network, with the second layer width $N_2$ fixed at 10. **a)** Second descent as a function of the first layer width $N_1$. **b)** Second descent as a function of the number of epochs, for $N_1 = 10$ and $N_1 = 100$.

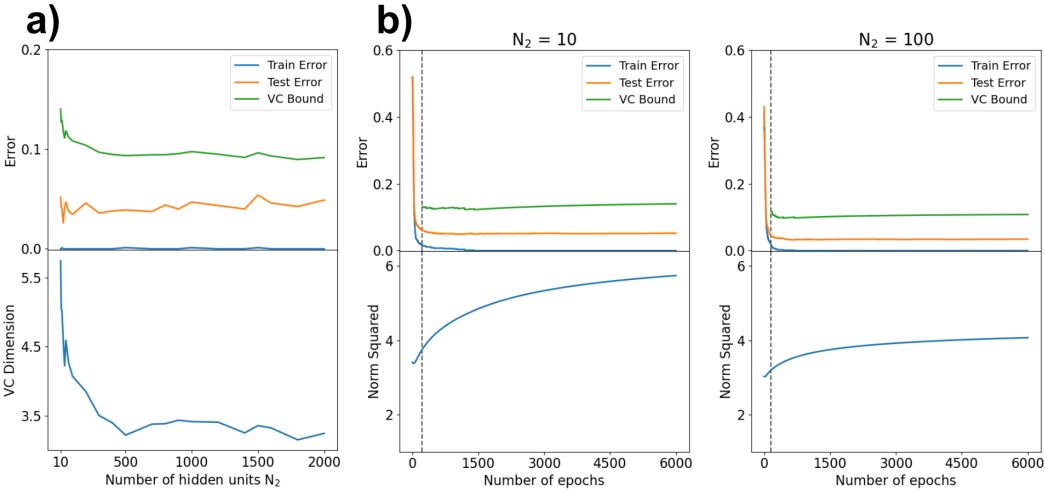

Figure 14: VC-theoretical modeling of second descent in a two-layer fully connected network, with the first layer width $N_1$ fixed at 10. **a)** Second descent as a function of the second layer width $N_2$. **b)** Second descent as a function of the number of epochs, for $N_2 = 10$ and $N_2 = 100$.

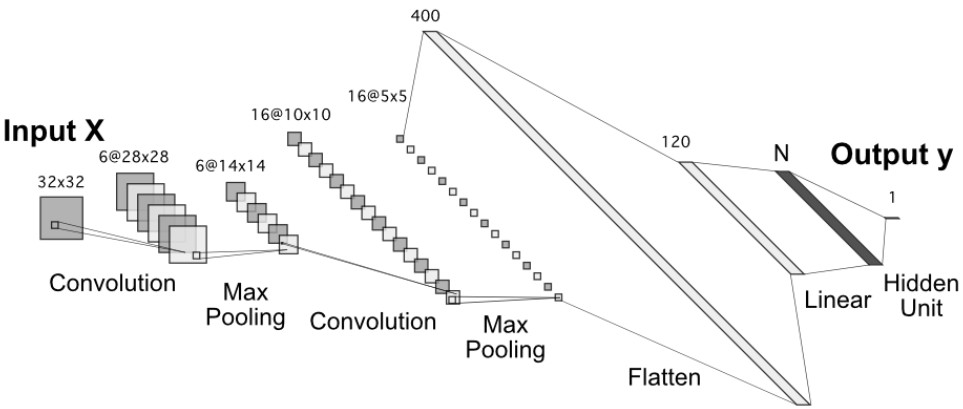

Figure 15: Schematic of LeNet-5 architecture adapted for binary classification setting, where the number of output units is set to 1. The size of the last hidden layer N is varied.

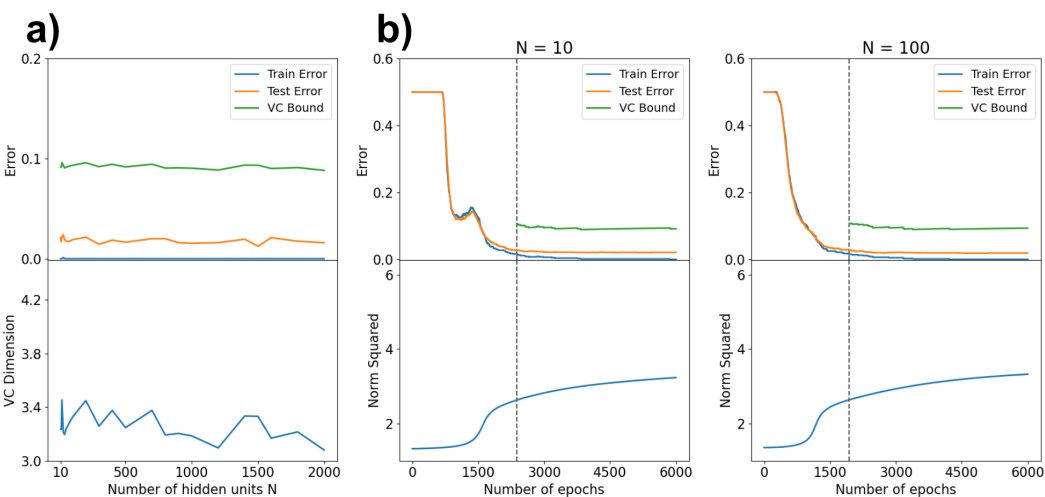

Figure 16: VC-theoretical modeling of second descent for LeNet-5. **a)** Second descent as a function of the last layer width N. **b)** Second descent as a function of the number of epochs, for N = 10 and N = 100.

