# OpenReview forum: "VC Theoretical Explanation of Double Descent"
_ICLR.cc/2023/Conference — Submitted to ICLR 2023_

### Official Review · Reviewer_Uf6C · 2022-10-17

**Confidence:** 4
**Correctness:** 1
**Technical Novelty And Significance:** 3
**Empirical Novelty And Significance:** 3
**Recommendation:** 3

**Clarity, Quality, Novelty And Reproducibility:**

[clarity] Most discussions are presented clearly. However, the main theory lacks many key details, making the correctness of the paper not judgable.

[novelty] The empirical coincidence of the curves of weight norm square and double desent for linear model is novel.

[reproducibility] The experimental plots are considered reproducible.

**Strength And Weaknesses:**

Strength:

The strength of the paper lies in that it empirically observes the perfect match of the curve of weight norm square and the double descent curve on several linear models. This suggests a deep connection between the weight norm square and the test error.

Weaknesses:

1: The paper only presents a few key formulas about the VC theory, but provides no theoretical derivation except a few references of prior works. As the VC-theory is the key of the paper, the authors should include a full analysis of all the details of the theory. The theory presented in the paper should be self-contained, such that readers with some background can understand the analysis/theory with minor or no reference to literature.

Without giving details of the VC-theory, I cannot judge the correctness of the paper.

2: Explaining double descent includes two parts: 1, theoretically connecting test error with weight norm square, or VC-dimension; and 2, theoretically explaining the behavior of weight norm square and VC-dimension.
The paper does not theoretically explain the behavior of the weight norm square. Even assuming all the formulas in the paper are correct, one still cannot claim the double descent is explained.

3: The formula of VC-dimension, in Eq.(3), only applies to linear models. However, the most interesting part of double descent is for non-linear models, e.g., over-parameterized neural networks. For the non-linear models, there is still no explanation.

4: The values of $a_1$ and $a_2$ are not from theoretical analysis, but from empirical guesses. Then, I don’t think the key equation, Eq.(2), is theoretical.


**Summary Of The Paper:**

This paper presents a formula bounding the test error using VC-dimension. By further considering linear models, the paper connects VC-dimension with the squared norm of weights, Eq.(3). The paper then empirically computes the weight norm square and observed that, for the linear models, the curve of weight norm square matches the double descent curve.

**Summary Of The Review:**

The observation of the empirical coincidence of the curves of weight norm square and double desent for linear model is interesting and encouraging. However, the main VC-theory is absent in the paper, and the behavior of weight norm square is not theoretically explained.

---

> ### Author Response · Authors · 2022-11-12
> **Response to Reviewer Uf6C**
>
> We thank the reviewer for reviewing our paper. Please see below our responses.
>
> **Regarding Point 1**
>
> We disagree with this criticism, because:
>
> - VC theory with proofs is described in Vapnik’s books provided in references. It is certainly not possible for us to copy these proofs into a conference paper
> - main VC theoretical concepts and results (without proofs) can be found in [Cherkassky & Mulier, 2007], that also presents conceptual interpretation of most existing machine learning methods using VC theoretical framework.
> - two theoretical results actually used in the paper, i.e. VC bound on test error and bound on VC dimension, are clearly presented and explained.
>
> It is not clear why this reviewer questions the validity of well-known fundamental VC-theoretical results – that appear in all of Vapnik’s books and many other textbooks.  This is methodological paper targeted to practitioners and researchers (such as ourselves) who believe that good engineering should be based on sound scientific theory. However, we are not focused on developing new theories and mathematical proofs.
>
> **Regarding Point 2**
>
> Theoretical explanation of the weights norm squared curve (for a linear network), as a function of N, the number of random nonlinear features, is given in [Belkin et al, 2018; Liang and Recht, 2021]. These papers show that minimum norm solution for least squares classifier, for large N, asymptotically approaches optimal solution provided by kernel method. The goal of our paper was to explain these (known) results using classical VC theory.
>
> **Regarding Point 3**
>
> We apply closed-form VC analytic bounds to a linear setting (with randomly generated features) previously used in  [Belkin et al, 2018, 2020, 2021; Liang and Recht, 2021] and multiple other papers trying to explain the mystery of double descent in DL. Methodologically, it stands to reason that theoretical understanding of double descent should be (initially) achieved for simple settings, and then proceed to more complex ones.
>
> It may not be possible to find closed-form analytic bounds for general nonlinear estimators such as DL nets, due to combined effect of complex DL architectures and various heuristics used for SGD training. Yet, there may be simple analytic formulas for VC dimension under restricted settings for DL. Our paper describes such restricted scenarios during second descent – along with empirical results for wide fully connected network with a single hidden layer in Section 3 and also additional results in Appendix B.
>
> For general nonlinear DL networks, it may be still possible to apply VC bound on test error assuming that VC dimension (of a trained DL model) can be *empirically estimated*. The procedure for empirical estimation of VC dimension can be used for *any nonlinear classifier*, as explained in [Vapnik, 2006; Cherkassky and Mulier, 2007]. We will mention this possibility in the revised paper, along with additional references.
>
> **Regarding Point 4**
>
> This is a rather disparaging remark about analytic bounds that apply to all possible finite data from unknown distributions and to all learning methods. As noted in the paper, VC theory provides the worst-values for these constants – but we suggest using specific value for all practical data sets. The fact that these practical values work well for various data sets shown in the paper is an accomplishment, rather than deficiency.
>
> **Reference**
>
> Vladimir Cherkassky and Filip Mulier. *Learning from Data*. Second Edition, Wiley-Interscience, 2007.
>
> Mikhail Belkin, Siyuan Ma, and Soumik Mandal. To understand deep learning we need to understand
> kernel learning. In *International Conference on Machine Learning*, pp. 541–549, 2018.
>
> Mikhail Belkin, Daniel Hsu, and Ji Xu. Two models of double descent for weak features. SIAM
> *Journal on Mathematics of Data Science*, 2(4), 2020.
>
> Mikhail Belkin. Fit without fear: Remarkable mathematical phenomena of deep learning through
> the prism of interpolation. *Acta Numerica*, 30:203–248, 2021.
>
> Tengyuan Liang, and Benjamin Recht. Interpolating classifiers make few mistakes. arXiv preprint arXiv:2101.11815v2, 2021.
>
> Vladimir Vapnik. *Estimation of Dependencies Based on Empirical Data*. Second Edition, Springer,  2006

---

> > ### Comment · Reviewer_Uf6C · 2022-11-17
> > **post-rebuttal comments**
> >
> > I thank the authors for giving feedbacks. However, most of my concerns stand.
> >
> > For point 1: I think it is necessary to write down the theoretical argument. Even if the major argument is from previous works, it is the authors' responsibility to present the whole theoretical argument, such that 1) reviewers and readers are clear whether the setting and assumptions in the current submission are consistent with the previous works, 2) reviewers and readers to fully understand, and justify the correctness of, the argument. Simply citing the related works does not give readers the detailed argument, as there are too much unrelated stuff in the books and papers.
> >
> > For point 2 & 3: The weight norm square curve for non-linear problems are still not theoretically known. Hence, to claim that VC theory explains double descent, the authors have to theoretically explain the weight norm square curve for non-linear-regression problems, especially for neural networks. Without a theoretical explanation of the weight norm square, I consider this work as only connecting the test error with weight norm square (assuming the correctness of the theory, although which is absent as mentioned in my point 1).
> >
> > For point 4: Given that the values of $a_1$ and $a_2$ are from experimental guesses, I don't think the VC theory in this paper is well-established, even if assuming the following arguments are correct.
> >
> > As my concerns stand, I keep my score.

---

> > > ### Author Response · Authors · 2022-11-19
> > > **Response**
> > >
> > > Thank you for your response. Please see our comments below.
> > >
> > > **For point 1**
> > >
> > > As clearly stated in the paper, we apply classical VC bounds under standard assumptions used in machine learning, similar to multiple publications [Zhang et al, 2017; Nakkiran et al, 2021] that also refer to VC bounds and VC-dimension when arguing that VC theory cannot explain the mystery of double descent. *None* of these published papers provides proof of VC bounds or elaborates on any of assumptions, as they are well known in the field. Our paper does not derive any new theoretical results, or propose new theoretical arguments, but simply explains the current confusion caused by misinterpretation of VC theoretical results in DL literature.
> > >
> > > **For point 2 & 3**
> > >
> > > It is not possible to explain analytically the weight norm_squared curve for general DL networks, due to inherent difficulty of quantifying the effect of multiple heuristic tuning parameters. However, we did explain the monotonically decreasing shape of the weights curve for linear networks with random nonlinear features.
> > >
> > > **For point 4**
> > >
> > > As explained in our paper, VC theory provides specific range of values for theoretical constants, and uses the worst-case values (from this range) for *qualitative conceptual explanation* of generalization (including double descent). Our paper suggests specific fixed values (from this range) that can be used for *quantitative modeling* of double descent, for most practical data sets. It is not clear why this reviewer is so opposed to the idea of using fixed practical values for theoretical constants – especially in view of the fact that all practical learning algorithms in DL contain dozens of heuristics with multiple tuning parameters (which are often tuned to a particular data set).
> > >
> > > **Reference**
> > >
> > > Chiyuan Zhang, Samy Bengio, Moritz Hardt, Benjamin Recht, and Oriol Vinyals. Understanding deep learning requires rethinking generalization. *In Proceedings of the International Conference on Learning Representation*, 2017.
> > >
> > > Preetum Nakkiran, Gal Kaplun, Yamini Bansal, Tristan Yang, Boaz Barak, and Ilya Sutskever. Deep double descent: Where bigger models and more data hurt. *Journal of Statistical Mechanics: Theory and Experiment*, 2021(12), 2021.

---

### Official Review · Reviewer_7rdX · 2022-10-20

**Confidence:** 4
**Correctness:** 4
**Technical Novelty And Significance:** 1
**Empirical Novelty And Significance:** 1
**Recommendation:** 3

**Clarity, Quality, Novelty And Reproducibility:**

The paper is well-written. However, as explained above, the main contribution of this article is not original.

**Strength And Weaknesses:**

Strengths:

- The paper is well-written and clearly makes a simple point regarding VC-dimension of neural networks and random feature models with L_2-bounded final layer weights

- I agree with the authors that in some prior literature there are overly strong statements about how the extent to which VC-dimension bounds are useful in the overparameterized regime. So it is a valuable message to push back again this point of view to some extent.

Weaknesses:
- The main point raised in this paper: namely that VC dimension with norm bounds on network weights can explain generalization in deep networks is not new. In my view, before this article can be published, it is imperative that a thorough review of and comparison with prior literature is undertaken. Once this is done, I am not certain whether there will be anything truly novel about the article under consideration. A short list of some relevant prior articles includes:
    - Bartlett’s paper [1] that is among the earliest works directly about one hidden layer neural networks with bounds on the norm of the weights in second layer the giving rise to VC-dimension bounds. This article has over 1700 citations.
    - Neyshabur et. al.’s work [2]. They consider in Theorems 1 and 2 rademacher complexity neural networks with bounds on the distance weights changed from initialization. They also review in Table 1 a range of other norm-based complexity measures that can given VC or Rademacher bounds. This article has over 400 citations.
    - Bartlett et. al.’s article [3], which considers Rademacher complexity bounds on overparameterized neural networks with a variety of norms constraints on all layer weights (see e.g. Theorem 3.3). This article has over 800 citations. This paper was already cited in the article under review. However, its use of weight norms to compute bounds on covering numbers is not discussed.

- The authors give the following as an example of an improperly held belief about the role of VC dimension in explaining generalization in overparameterized neural networks: “Another common view is that “VC-dimension depends only on the model family and data distribution, and not on the training procedure used to find models” (Nakkiran et al., 2021). In fact, VC-dimension does not depend on data distribution.” I believe this assertion deserves a more nuanced treatment. Specifically, VC-dimension bounds on norm-constrained classes of predictors can only really be useful when the size of the norm-constraints depends on the data-distribution! For instance, consider a random feature model $f(x) = \theta \cdot x$ with a bound $||\theta||_2 \leq B$ on the model weights. If $y(x)$ is, say $\pm 1$,  independent of x, then B will have to be much larger in order to fit the data and give non-vaucuous bounds, compared with the case when $E[y(x) | x]$ is a say a smooth function. Hence, this seems like an important example where VC-dimension should depend on the data distribution (via the choice of a priori constrains on the collection of functions whose VC-dimension is being computed).


[1] Bartlett, Peter. "For valid generalization the size of the weights is more important than the size of the network." Advances in neural information processing systems 9 (1996).

[2] Neyshabur, Behnam, et al. "Towards understanding the role of over-parametrization in generalization of neural networks." arXiv preprint arXiv:1805.12076 (2018).

[3] Bartlett, Peter L., Dylan J. Foster, and Matus J. Telgarsky. "Spectrally-normalized margin bounds for neural networks." Advances in neural information processing systems 30 (2017).

**Summary Of The Paper:**

This article makes the case that the double descent phenomenon in overparameterized neural networks and random feature models can be explained by considering the VC-dimension of spaces of such functions with L_2-bounded final layer weights. Moreover, the authors claim that this point has been missed in the existing literature.

**Summary Of The Review:**

This article proposes that VC-dimension bounds of classes of neural networks with bounds on the norm of the final layer weights can explain double descent. However, the article fails to differentiate its point of view from the majority of prior literature on this subject. As a result, I believe this article is not ready for publication.

---

> ### Author Response · Authors · 2022-11-12
> **Response to Reviewer 7rdX**
>
> Thank you for reviewing our paper. Please see our response below.
>
> **Regarding Point 1**
>
> We agree that the paper was missing several important references – they will be added in the revised version.
>
> However, we strongly disagree with suggestion that there is nothing new in the paper. Our main point is not about using the norm of weights (narrow technical result), but a general methodological message that VC theory can fully account for generalization in DL (along with other learning methods).  Further, our main technical result is practical application of VC bounds. Classical VC bounds *have never been used before* for quantitative modeling of test error in general and double descent, in particular. This certainly has major importance for many researchers and practitioners. Classical VC theoretical bounds have been introduced for conceptual understanding of generalization. They contain various theoretical constants that reflect general properties of unknown distributions – so that using worst-case values of these constants makes these bounds practically useless (i.e., too conservative). See, for example, [Hastie et al, 2009; Vapnik, 1998; Cherkassky & Mulier, 2007]. Our paper demonstrates that VC bounds, with properly chosen values for theoretical constants, can be used for quantitative modeling of double descent. Notably, the *same values* of theoretical constants have been used for various data sets and for different classifiers.
>
> In comparing our approach with other work using norm-based capacity indices, it is also worth noting that we use general VC bounds that have been derived before DL and double descent became known. In contrast, many related papers propose norm-based indices in an attempt to explain empirically observed double descent phenomenon.
>
> **Regarding the quote from our paper**
>
> We believe that misunderstanding is caused by using the term ‘VC dimension’ in two different contexts. That is, our original sentence (quoted above) refers to VC dimension of a set of functions, referring to VC theoretical structure, defined a priori (independent of data distribution). This is the context in which VC dimension is mentioned in the paper by [Nakkiran et al, 2021]. In contrast, Reviewer’s comment refers to VC dimension of an optimal element of a structure selected using particular training data (that, of course, depends on data distribution). In summary, a structure is defined a priori, but an optimal element of a structure is found using training data (so, it is data-dependent).
>
> Using the notion of VC dimension for understanding practical learning methods, such as DL or SVM, always involves the concept of VC theoretical structure, or complexity ordering, specified a priori on a set of admissible models. This concept of a structure appears to be misunderstood in DL.
>
> **Regarding the remark on VC dimension should depend on data distribution**
>
> Yes, norm-based VC dimension additionally assumes that all input (x) samples come from a unit sphere. This constraint is clearly stated in our paper and in all quoted references on VC theory. In practical DL networks, this condition is probably enforced by various heuristics, such as weight initialization and batch normalization in each layer.
>
> Interpreting this condition as the ‘dependency’ of VC dimension on data distribution may be a matter of personal choice. But it certainly cannot be interpreted as the dependency of VC dimension on (unknown) class distributions.
>
> **Reference**
>
> T. Hastie, J. Friedman and R. Tibshirani, *The Elements of Statistical Learning*, Springer 2009
>
> Vladimir Vapnik. *Statistical Learning Theory*. John Wiley & Sons, New York, 1998.
>
> Vladimir Cherkassky and Filip Mulier. *Learning from Data*. Second Edition, Wiley-Interscience, 2007.
>
> Preetum Nakkiran, Gal Kaplun, Yamini Bansal, Tristan Yang, Boaz Barak, and Ilya Sutskever. Deep double descent: Where bigger models and more data hurt. *Journal of Statistical Mechanics: Theory and Experiment*, 2021(12), 2021.

---

### Official Review · Reviewer_J8jr · 2022-10-21

**Confidence:** 3
**Correctness:** 2
**Technical Novelty And Significance:** 2
**Empirical Novelty And Significance:** 2
**Recommendation:** 3

**Clarity, Quality, Novelty And Reproducibility:**

The paper is clearly written and the experiments are described enough to be reproducible.

**Strength And Weaknesses:**

The goal of the paper is certainly commendable; revisiting classical ideas may shed insights into recent empirical observations.

However, I don’t think that the authors apply VC theory correctly in their argument. First, it is important to keep in mind that one does *not* minimize a VC dimension because the VC dimension is a property of the hypothesis space that you consider during training. The only valid way to reduce the VC dimension is to change the hypothesis space *from the outset*, not during training. It seems that the authors allude to structural risk minimization (SRM), but SRM is different. In SRM, one has a nested set of hypothesis spaces and learning proceeds from the smallest to the largest hypothesis spaces. In this case, the VC dimension is the VC dimension of the largest set ever considered. Note that all hypotheses outside of the last set in SRM are *never* considered during training. In the arguments the authors make, one starts with a complex hypothesis space and then try to reduce it by picking a smaller set if any. This is not a correct application of VC theory.

Second, while authors focus on linear classification, the extension of their argument to deeper architectures is based on an unsupported hypothesis without any proof. The authors hypothesize that one can treat a deep neural network as a linear classifier on the pre-logit features and then proceed by stating that the norm of the weights on the classifier’s head gives a measure of generalization. One problem with this argument is that when using activations like ReLU, we can always rescale the weights of the early layers and decrease the norm of the weights of the classifier head without changing the decision boundary. That's why in norm-based generalization bounds, such as in [1, 2, 3, 4], it is the *product* of the norms of all layers that count. In addition, since the product itself can be increased or decreased arbitrarily without changing the decision boundary, generalization bounds almost always normalize the product of the norms of the layers by the *margin* on the training examples.

Third, the VC dimension is independent of the distribution of the data. If there is a claim that neural networks have a small VC dimension, then neural networks should generalize across *all* data distributions including those with random labels. But, clearly that’s not the case.

[1] Bartlett, P. L. The sample complexity of pattern classification with neural networks: the size of the weights is more
important than the size of the network. IEEE Transactions on Information Theory, 44(2): 525–536, 1998.

[2] Bartlett, P. L.; Foster, D. J.; and Telgarsky, M. J. Spectrally-normalized margin bounds for neural networks. NeurIPS, 2017

[3] Arora, S.; Ge, R.; Neyshabur, B.; and Zhang, Y.  Stronger generalization bounds for deep nets via a compression approach. ICML, 2018.

[4] Neyshabur, B.; Tomioka, R.; and Srebro, N. Normbased capacity control in neural networks. COLT, 2015.


**Summary Of The Paper:**

The paper aims to provide an explanation of double descent using classical VC theory. The primary argument is that since the test error can be bounded by a function involving both the training error and the VC dimension, one may observe double descent if one minimizes the training error and then “minimizes the VC dimension” subject to having a small training error (e.g. zero). The authors argue that this explains double descent in modern neural networks. They conduct experiments in simple settings (mostly linear classification) to support their argument.


**Summary Of The Review:**

I do not believe VC theory is applied correctly in this paper. Also, the main hypothesis that the authors postulate for deep neural networks is not valid in my opinion since one can rescale the weights in the early layers and reduce the norm of the weights at the last layer without changing the decision boundary.

---

> ### Author Response · Authors · 2022-11-12
> **Response to Reviewer J8jr**
>
> Thank you for reviewing our paper. Please see below our responses.
>
> **Regarding Point 1**
>
> With all due respect, Reviewer’s assertions are *not correct*. As clearly explained in our paper, there are two distinctly different SRM strategies. Under Strategy 2, implemented during second descent, we consider a structure formed by a set of linear models, ordered by constraint on the norm of weights. This structure is defined *a priori*, and it *does not* depend on training data. However, since we have an over-parameterized setting, every element of this structure will interpolate training data, i.e. every element will have zero training error. An optimal element will have the smallest norm, i.e. the smallest VC dimension, for a given training set. The same strategy has been implemented in standard (nonlinear) SVM – where minimum norm solution is found using training data. For most practical applications, nonlinear SVM models implement hard-margin SVM in some infinite-dimensional space induced by nonlinear kernel – therefore implementing *exactly* Strategy 2. This is well-known and described in Vapnik’s books and other references on SVM aka kernel methods. See, for example, Section 10.7 in [Vapnik, 1998], *describing exactly* how this SRM strategy works for selecting the best SVM model corresponding to largest-margin for training data. Direct quotation from Section 10.7: ‘Note that the SVMs implement the SRM principle’.
>
> Results shown in our paper demonstrate that exactly the same mechanism and the same VC-bound work well for controlling VC dimension of (linear) classifiers estimated via SGD using squared loss. Similar findings were reported by other researchers in DL, who re-discovered this obvious similarity between SVM and DL (used in second descent mode). See, for example, [Belkin et al., 2018; Liang et al., 2021]. However, many of these papers start with a premise that VC theory does not work for DL, and then proceed to discover that generalization in DL is controlled by the norm of weights. Their view is self-contradictory, because the notions of margin and SVM method have been theoretically justified in VC theory, using VC bounds and SRM.
>
> **Regarding Point 2**
>
> In our paper, this method is not presented as a general approach – we simply hypothesized that it can work (in conjunction with VC bounds) for *well-defined DL settings* in second descent mode, as confirmed by empirical results. It is important to note here, that VC dimension of DL networks is *highly dependent* on various heuristics used for SGD training. Therefore, *all theoretical estimates* of VC dimension, including the spectral norm, advocated by the Reviewer, are *probably flawed*, since they may not account for the effect of such heuristics in practical applications.
>
> Therefore, all empirical results for DL and CNN, reported in our paper in Appendix B, are based on using *existing* DL architectures (such as CNN for digits) along with existing SGD implementations. In other words, we did not attempt additional tuning of hyper-parameters of SGD, but used standard settings taken from other papers.
>
> Responding to reviewer’s concern about arbitrary normalization in each layer we note that:
> - our empirical results for DL networks use existing SGD implementations (from other studies) that include standard ‘batch normalization’ during training. We believe that batch normalization tends to approximate the condition that the output values of all units (in a layer) are inside a unit sphere. This condition is needed for radius-margin bound on VC dimension, shown as equation (3) in our paper.
> - for empirical validation, we repeated experiments reported in Appendix B, but without using batch normalization. These new results suggest that removing batch normalization negatively affects modeling results shown in Appendix B, by making VC bound estimates of empirical test error curves much less accurate.
>
> **Regarding Point 3**
>
> We do not quite understand this criticism. However, our paper includes empirical results for modeling digits data with randomly corrupted labels (see Fig. 3) – they clearly show that, in second descent mode, a classifier trained to memorize noisy data has larger norm of weights than a classifier trained using the same digits but without label noise. This is in perfect agreement with our understanding of VC theoretical bounds.
>
> **Reference**
>
> Vladimir Vapnik. *Statistical Learning Theory*. John Wiley & Sons, New York, 1998.
>
> Mikhail Belkin, Siyuan Ma, and Soumik Mandal. To understand deep learning we need to understand
> kernel learning. In *International Conference on Machine Learning*, pp. 541–549, 2018.
>
> Tengyuan Liang, and Benjamin Recht. Interpolating classifiers make few mistakes. *arXiv preprint arXiv:2101.11815v2*, 2021.

---

> > ### Comment · Reviewer_J8jr · 2022-11-14
> > **Thanks for the clarification**
> >
> > Thank you for the response. I see your point regarding my first objection; I think the primary confusion comes from the wording. For example, you mention explicitly that in the second descent: "for small (fixed) training error, minimize the VC-dimension." This is confusing because you cannot fix the training error without defining the hypothesis space *according* to the training data.
> >
> > But, l understand your view now; the argument is that small-norm solutions are sufficient to interpolate the data and one can fix the norm in advance so we can speak about the VC dimension for those nest hypothesis spaces with increasing norms. For random labels, a small-norm solution does not interpolate (as you observe empirically), so there is no contradiction.
> >
> > However, this does not really answer my main criticism. The central claim of the paper is that VC theory *can explain* double descent and generalization in neural networks. But then, the explanation is based on a hypothesis that is not well-supported. I gave an example about rescaling and you said that you assume batch-normalization but where does batch-normalization fit into the theoretical framework? I know you used them in the experiments but how do they fit in the theory, especially given that batch normalization does not really fix anything since the batchnorm layer contains a scale parameter $\gamma$ and one can multiply by $\gamma$ and divide the weight in the next layer by the same constant without affecting the decision boundary (assuming no bias/offset terms).
> >
> > I don't think an argument based on the norm alone will give a meaningful bound for the reasons I mentioned earlier. For this, most of the previous bounds on generalization normalize the norm by a margin on the training data.
> >
> > In any case, I'm going to update my score.

---

> > > ### Author Response · Authors · 2022-11-15
> > > **Re: Response to Reviewer J8jr**
> > >
> > > We are glad that we clarified your main concern regarding application of VC bounds in second descent mode (when training error is zero).
> > >
> > > With regard to your remaining concern that the paper does not show that VC theory can explain double descent:
> > >
> > > The main claim of the paper is that VC bounds can explain double descent – this is well explained on a conceptual/methodological level, and also supported by empirical results for *restricted settings* when VC-dimension can be analytically estimated. Both our conceptual explanation and quantitative modeling results are new and contradict the consensus view in the Deep Learning community. The restricted setting with nonlinear random features (used in our paper) has been adapted from [Neyshabur et al, 2015; Belkin et al, 2018] trying to explain generalization in Deep Learning. For this setting, we showed that VC bounds can model double descent very accurately, and the resulting double descent curves have meaningful VC theoretical interpretation.
> > >
> > > For ‘true’ multilayer networks, we do not have an accurate estimate of VC dimension, and therefore cannot apply VC bounds for quantitative modeling. This is a technical difficulty and does not prevent conceptual understanding using VC bounds. We hypothesize that under *certain restricted settings*, during second descent mode, it is possible to use the norm of weights in the last layer, as a proxy for VC-dimension. This hypothesis is supported by limited empirical results for several well-defined DL architectures. These results were obtained by using SGD training regimes taken from other DL papers, i.e. we did not attempt to tune various DL heuristics in order to improve our modeling results. In any case, we present these results as a hypothesis, not as a general theoretical result - and will state so in the revised paper.
> > >
> > > The reviewer comment seems to reflect an expectation that a true theoretical explanation of double descent should be able to incorporate directly the effect of various DL heuristics (such as batch normalization, weight initialization etc.) into analytic bounds. We do not think this is realistically possible to accomplish. Yet, it may be possible to measure the VC dimension of a well-trained DL model, assuming that VC dimension can be *empirically evaluated*. The method for empirical measuring of VC dimension (of any nonlinear classifier) is described in [Vapnik, 2006; Cherkassky and Mulier, 2007]. This method is based on an intuitive observation that an estimator with large VC dimension (h) is more likely to overfit a finite data set (of size n) with randomly chosen class labels. The deviation of observed training error from 0.5, as a function of n, can be quantified in VC theory, providing the basis for the method. We will mention this possibility in the revised paper, along with additional references.
> > >
> > > In response to technical comment about the scaling parameter in batch normalization, note that batch normalization effectively controls the size of the enclosing sphere used to control the VC dimension via equation (3). While the two parameters in batch normalization (scaling parameter γ and bias β) are tunable during training, we found that their values are very close to γ=1 and β=0 in the end of training, for all trained models (with different number of hidden units N). In fact, if all neural network models in section 3 of the paper are trained with *fixed values* γ and β set to 1 and 0 respectively, then all modeling results remain unchanged. This indicates that data-dependent tuning of these parameters has no effect on generalization curves, at least for networks and data sets used in the paper. Since the Z values are normalized, any scaling that occurred in the previous layers would not affect the last layer weights. We will include justification for using batch normalization in the revised paper.
> > >
> > > **References**
> > >
> > > Mikhail Belkin, Daniel Hsu, Siyuan Ma, and Soumik Mandal. Reconciling modern machine learning practice and the classical bias–variance trade-off. Proceedings of the National Academy of Sciences, 116(32):15849–15854, 2019.
> > >
> > > Behnam Neyshabur, Ryota Tomioka, and Nathan Srebro. In search of the real inductive bias: On the role of implicit regularization in deep learning. Proceeding of the International Conference on Learning Representations workshop track, 2015.
> > >
> > > Vladimir Vapnik. Estimation of Dependencies Based on Empirical Data. Second Edition, Springer, 2006.
> > >
> > > Vladimir Cherkassky and Filip Mulier. Learning from Data. Second Edition, Wiley-Interscience, 2007.

---

> > > > ### Comment · Reviewer_J8jr · 2022-11-17
> > > > **Response**
> > > >
> > > > Thanks for the response.
> > > >
> > > > I think we have a different meaning of what "explain" means. I don't think you have shown that VC theory explains double descent. What you argue for is a conjecture/hypothesis that is inspired by VC theory, not an actual explanation. In particular, if there was no double descent, nothing in your analysis would predict it. An explanation should prove that double descent arise necessarily (perhaps under certain conditions).
> > > >
> > > > My objection is that the paper is based on two hypotheses/speculations. First, you speculate that training goes through two phases: (1) minimize the training error, and (2) minimize the norm subject to the training error. You argue that these two phases correspond to two descents. But, if it's a matter of presenting a conceptual understanding, one can instead just "claim" that training goes through two phases: (1) minimize the training error, and (2) minimize the generalization gap subject to zero training error. But then, this statement is not different from previous works, such as (Shwartz-Ziv and Tishby, 2017) who observed empirically that neural networks goes through two phases: fitting followed by compression.
> > > >
> > > > Second, you speculate that the norm of the last layer is enough. As I mentioned before, this is not a valid statement. I'm not saying that one should take into account factors such as batchnormalization. What I'm saying is that the norm of the last layer alone cannot provide a meaningful bound for the reasons I mentioned above.
> > > >
> > > > P.S. Where does the ascent (in the middle) come from according to your argument? Otherwise, we would have one descent.

---

> > > > > ### Author Response · Authors · 2022-11-19
> > > > > **Response**
> > > > >
> > > > > Thank you for your response. Please see our comments below.
> > > > >
> > > > > **About the VC Theory explain double descent**
> > > > >
> > > > > We use the term ‘explanation’ to indicate that VC-theory (developed long before DL and double descent were known) can provide explanation for recently observed empirical phenomenon (~double descent). This is a common and proper use of the term ‘scientific explanation’, as it is used in classical science. In contrast, many recent theoretical explanations of double descent should be probably regarded as conjectures because they usually introduce new theoretical concepts for explaining this particular empirical phenomenon.
> > > > >
> > > > > According to VC-theoretical explanation, double descent displays application of two different VC-theoretical structures, or two different learning methods, applied to the same data set. Typically, a learning method implements a single structure, so there is no need for displaying the effect of using two different structures. For example, SVM methods are usually applied in second descent mode, whereas classical statistical methods implement first descent. In DL, ‘network size’ is viewed as a tunable parameter, which leads to confusion, because learning using small (under-parameterized) network and large (over-parameterized) network effectively implements two different learning algorithms.
> > > > >
> > > > > The reviewer refers to these two different structures as ‘training going through two phases’, but this is not what is stated in our paper. Again, we emphasize that VC-theoretical concepts and results are general, and they explain both (a) traditional methods implementing ‘first descent’ learning, (b) modern methods such as SVM implementing ‘second descent’. This is in contrast to multiple new theoretical explanations specifically designed for explanation of double descent.
> > > > >
> > > > > **About the last layer norm**
> > > > >
> > > > > We do present this assertion as a hypothesis that works under restricted settings – as outlined in the paper. These caveats are clearly stated in the paper. However, we also show empirical results indicating that this hypothesis works under many restricted settings. In summary, the disagreement arises because the Reviewer interprets our hypothesis as a theoretical assertion, which it is not.
> > > > >
> > > > > **About the ascent**
> > > > >
> > > > > There is actually no ascent – because double descent curve simply displays two different generalization curves: one for the first descent (classical bias-variance shape) and for second descent (modern regime). Both types of generalization curves are fully explained in VC theory, but they were presented separately, rather than displayed next to each other.

---

### Official Review · Reviewer_4NbT · 2022-10-24

**Confidence:** 3
**Correctness:** 3
**Technical Novelty And Significance:** 3
**Empirical Novelty And Significance:** 3
**Recommendation:** 6

**Clarity, Quality, Novelty And Reproducibility:**

The paper is generally quite well written.

I'm not familiar with the VC bounds in the form (1), (3) playing a central role in the paper, and the paper only vaguely refers to the books of Vapnik and Cherkassky & Mulier as the sources. It would be helpful to see more specific references or maybe an appendix with exact formulations of relevant theorems.


**Strength And Weaknesses:**

Strengths
-------------
The main strength of the paper is that it makes a clear point challenging a common belief, and consistently argues in favor of this point. The paper is quite thoughtfully written. The conceptual problem addressed - practically deriving the complex dependence of model performance on model complexity and sample size - is important and hard. The experimental demonstrations in the case of shallow networks look convincing to me. I think the paper has a definite methodological value and can be useful to the readers.

Weaknesses
----------------
The main weakness that I see is that the main claim of the paper, that theoretical VC bounds can be used to describe double descent, is essentially only demonstrated for shallow models or in effectively shallow settings, when it is not that interesting because there are now other analytical methods applicable in these settings that can often provide more detailed information (e.g., explicit solutions based on random matrix theory).
Also, I'm not an expert in statistical learning theory, and it's hard for me to judge, but the paper contains no new technical results; its novelty seems to be limited to interpreting old VC bounds and performing an empirical study in a new context. The key idea on which the discussion of double descent rests in the paper is that VC dimension is primarily determined by the number of parameters in the underparameterized regime but rather by the weight norm in the overparameterized regime. In some form, this or closely related ideas seem to have already been around for a while in recent works on double descent (and are in a sense already present in bound (3)). However, this does not mean that I doubt the novelty of specific details of theoretical analysis and experimental setup in the present paper.





**Summary Of The Paper:**

The paper argues that, contrary to a common belief, the "double descent" effect can be fully explained by classical VC-generalization bounds. The paper argues that this requires a proper form and interpretation of the bounds, which are discussed at length. The authors then perform an empirical study of several models on the MNIST data in which the VC bound curve can be plotted and shown to describe double descent.


**Summary Of The Review:**

A solid well-written paper offering a comprehensive discussion of the very important topic of explaining double descent and related effects via classical VC dimension theory. However, I'm not fully convinced in the significance of the results, due to a lack of any analytical novelty and a relatively limited scope of demonstrated applications.

---

> ### Author Response · Authors · 2022-11-12
> **Response to Reviewer 4NbT**
>
> We thank the reviewer for reviewing our paper. Please see below our responses:
>
> **Point 1: "In some form, this or closely related ideas seem to have already been around for a while in recent works on double descent (and are in a sense already present in bound (3))."**
>
> Yes, we agree that similar technical claims have been made in other recent papers. However, this observation simply re-confirms our main (methodological) message that VC theory can fully explain double descent phenomenon. So, there is no need to invent multiple new theoretical explanations. Often, such new theories tend to re-discover known VC theoretical results, but introduce various additional assumptions, new concepts and new terminology. In this regard, these theories are inconsistent (with each other), and simply add to existing confusion in the field.
>
> On the contrary, proposed VC framework immediately clarifies understanding of many issues related to double descent, as discussed in the paper.
>
>
>
> **Point 2: "It would be helpful to see more specific references or maybe an appendix with exact formulations of relevant theorems."**
>
> In revised paper, we will add specific pointers to book sections containing main VC theoretical results. However, at this point it may be appropriate to mention that VC theory provides a conceptual framework for machine learning, not just mathematical results – and this framework is very different from the one adopted in DL. Practical application of VC theoretical results, which is our main goal, requires understanding this framework. It is very true that original Vapnik’s books are hard to follow, even though his presentation is always very logical and self-consistent. For that reason, practitioners (not interested in math proofs) can find useful the textbook by [Cherkassky & Mulier, 2007] that contain description of machine learning under VC framework.
>
> Vladimir Cherkassky and Filip Mulier. *Learning from Data*. Second Edition, Wiley-Interscience, 2007.

---

### Author Response · Authors · 2022-11-12
**Response to All Reviewer Regarding Methodological Issues**

Our overall impression is that wide range of reviews from reflects misunderstanding of VC theory in DL. Some comments reflect reviewers’ expectations of new theoretical or empirical results. However, this is mainly a methodological paper. This response comments on methodological importance of VC theory for understanding generalization in DL.

To understand the current state of affairs, recall that DL has been driven mainly by practitioners focused on applications for Big Data. Practitioners explained success of DL by similarity to biological networks, or properties of gradient-descent learning, or multilayer topology. Such explanations lacked rigorous support, yet they formed a consensus about superiority of DL vs. ‘shallow’. This consensus view was later reinforced by observing that most DL models are over-parameterized, and yet can generalize well. This has led in another consensus view, that “understanding deep learning requires rethinking generalization”, suggesting that classical VC theory cannot explain generalization in DL. This view has been immediately adopted by mathematicians, proposing new theories for second descent. Currently, we have multiple new theoretical explanations, often inconsistent with each other.

VC theory provides *necessary* and *sufficient* conditions for generalization of all methods based on minimization of training error. Hence, there is a clear disagreement between VC theory and the consensus view in DL, that can be explained by:
- different underlying assumptions in VC theory and DL. This is not the case, as both assume the same distribution for training and test data.
- VC theory is incorrect. But there are no theoretical proofs that invalidate VC theory. Supporters of DL just argue that generalization of large networks cannot be explained in terms of classical bias-variance trade-off.
- misunderstanding of VC theoretical results in DL – as explained in our paper.

Our paper claims that classical VC theory can fully account for generalization of over-parameterized models. This is significant, because VC-theoretical concepts have been introduced independent of the empirical phenomenon (double descent), unlike multiple theories specifically designed for explanation of double descent. Since VC theory gives necessary and sufficient condition for generalization, any new theory (outside VC framework) is likely to re-discover VC results. This is evident with many emerging explanations, i.e.
- papers suggesting that generalization in DL is controlled paper by the norm of weights [Belkin et al 2018, Bartlett et al, 2017, Neyshabur et al, 2017, Neyshabur et al, 2019]. Belkin et al (2018), argues that large-size networks, trained using SGD, tend to converge to the minimum norm solution - similar to kernel methods. This is identical to conclusions made in our paper based on analysis of classical VC bounds. Yet, Belkin et al (2018) also state ‘While the margins theory can be used to study classification, it does not apply to regression, and also does not predict the second descent beyond the interpolation threshold’. This statement is false both technically and methodologically, since there is no separate margins theory, as SVMs have been invented within VC theory.

Methodologically, multiple new theories for double descent, not constrained by a general conceptual framework, will be tailored to specific ad hoc chosen notions of complexity and particular selection of tunable DL heuristics. So, they will not likely improve fundamental understanding of generalization in DL. We do not suggest that VC theory magically provides simple explanation for all heuristics in DL. However, understanding the effect of these heuristics on generalization can be improved by relating them to fundamental concepts, such as VC dimension and structure.

Under VC interpretation, common ‘double descent’ shape of test error reflects two different types of VC-structures, for the same training data set.  In VC theory, the choice of a particular structure cannot be theoretically justified. Typically, a learning method implements a single structure, so there is no need for displaying the effect of using two different structures. For example, SVM methods are usually applied in second descent mode, whereas classical statistical methods implement first descent.

Contrary to consensus opinion in DL that second descent always works better, VC theoretical analysis in our paper shows that for data with noisy labels first descent mode of learning may provide better generalization– in full agreement with VC bounds. Our paper also also invalidates another common belief, that generalization in DL networks trained on data with noisy labels cannot be explained by VC theory.

We see the value of our paper in challenging the consensus views based on misinterpretation of VC theory. Our paper presents both (a) qualitative VC theoretical explanation of double descent and (b) empirical modeling for restricted settings.

---

### Decision · Program_Chairs · 2023-01-20

**Decision:**

Reject

**Justification For Why Not Higher Score:**

As explained in my detailed meta-review, the paper makes a bold claim that it fails to support in a convincing way. Toning down the abstract and introduction, unfortunately, won't be enough to fix this issue. The authors should at least address three concrete points, raised by the reviewers (and listed in my meta-review), before the work can be considered for the publication.

**Justification For Why Not Lower Score:**

N/A

**Metareview: Summary, Strengths And Weaknesses:**

The main claim of the paper is that it "fully explains" double descent phenomenon in Deep Learning using VC theory. The authors claim to achieve this by applying VC theory and the corresponding bounds in a correct way. Significant part of the paper is the discussion around the fact that VC theory has largely been misunderstood by the Deep Learning community.

The authors build their suggested interpretation on two results from the VC theory: (a) the VC bound on the generalization gap that has empirical error as a multiplier in front of the classical $\sqrt{VC / n}$ term (Equation 1); and (b) the margin bound on the VC dimension (Equation 3). Both results are standard and well known in the literature: certainly in the statistical learning community and, perhaps to a lesser extent, in the Deep Learning community. After describing their interpretation, the authors demonstrate empirical results (for linear models) to support their claims.

I agree with the reviewers, who pointed out that revisiting old classical results in light of new developments can be an important and fruitful direction and that it "may shed insights into recent empirical observations". I also fully agree that "in some prior literature there are overly strong statements about ... extent to which VC-dimension bounds are useful in the overparameterized regime" and that it is important to "push back again[st] this point of view to some extent".

Unfortunately, after reading the paper and also based on my detailed account of the entire reviewing and rebuttal process I have to agree with the reviewers and conclude that at this point the authors failed to support their claim ("fully explain the double descent in Deep Learning") in a convincing way. It has been noticed that what the authors argue for is "a conjecture/hypothesis that is inspired by VC theory, not an actual explanation", that the "explanation of double descent in the paper is fairly weak", and that "the paper "demonstrates" rather than "explains" it". The reviewers also expressed a concern regarding the novelty and contribution of the paper, questioning "whether there will be anything truly novel about the article under consideration", noticing that the paper "contains no new technical results" and that rather "its novelty ... [is] limited to interpreting old VC bounds and performing an empirical study in a new context".

Considering all these points and given how bold the claims made in the paper are, I feel the paper can't be published in its current form.

There are several concerns raised by most of the reviewers.

First, while both the "formal" and empirical parts of the paper are focused on the linear setup, the authors extrapolate to the general Deep Learning (which is the main subject of the paper's claim) using a rather hand wavy argument. Quoting the reviewers, "the extension of their argument to deeper architectures is based on an unsupported hypothesis without any proof". It has been also noticed that the margin based VC-bound in Equation 3 "only applies to linear models" and that "for the non-linear models, there is still no explanation".

Second, the formalism within the linear setup also raised some concerns. Very likely the authors can improve on this point by providing detailed and clear definitions, whenever applicable. The reviewers noticed that the paper is"very unspecific in citing the references for many of its general statements" and that it is "authors' responsibility to present the whole theoretical argument" in a clear, formal, and transparent way. "Simply citing the related works does not give readers the detailed argument". For example, I would suggest the authors to write down the precise definition of VC dimension *before presenting their proposed explanation* and to refer to it whenever making any claims related to the VC dimension. Equation 3 should appear together with all the accompanying assumptions (that the class only consists of predictors perfectly separating a given set with a required margin etc) made explicit. Also, I would suggest the authors to set in stone what exactly they mean by the "double descent phenomenon" --- is it with respect to the training step, or the number of parameters, or the model capacity? All versions have been presented and studied in the literature.

Third, the reviewers agree that the paper is largely missing a detailed discussion of the related results, including more recent works on the bounds accounting for various weight norms. They agreed that "it is imperative that a thorough review of and comparison with prior literature is undertaken".

Finally, if I am allowed to do so, I want to suggest the authors to be more forgiving and restrained in future, especially during the rebuttal phase.

**Summary Of Ac-Reviewer Meeting:**

N/A